# Synthesis of Rigid Polyurethane Foams Incorporating Polyols from Chemical Recycling of Post-Industrial Waste Polyurethane Foams

**DOI:** 10.3390/polym14061157

**Published:** 2022-03-14

**Authors:** Izotz Amundarain, Rafael Miguel-Fernández, Asier Asueta, Sara García-Fernández, Sixto Arnaiz

**Affiliations:** GAIKER Technology Centre, Basque Research and Technology Alliance (BRTA), Parque Tecnológico de Bizkaia, Edificio 202, 48170 Zamudio, Spain; miguel@gaiker.es (R.M.-F.); asueta@gaiker.es (A.A.); garciafdezsara@gmail.com (S.G.-F.); arnaiz@gaiker.es (S.A.)

**Keywords:** glycolysis, polyurethane, foams, chemical recycling, circular economy, polyols

## Abstract

The preparation and characteristics of rigid polyurethane foams (RPUFs) synthesized from polyols obtained by glycolysis of post-industrial waste RPUFs have been studied. More precisely, waste rigid foams that have been chemically recycled by glycolysis in this work are industrially produced pieces for housing and bracket applications. The glycolysis products have been purified by vacuum distillation. The physicochemical properties of the polyols, such as hydroxyl value, acid value, average molecular weight (M_n_) and viscosity have been analyzed. The chemical structure and thermal stability of the polyols have been studied by means of FTIR and TGA, respectively. Partial substitution of the commercial polyol (up to 15 wt.%) by the recycled polyols increases the reactivity of the RPUFs synthesis, according to short characteristic times during the foaming process along with more exothermic temperature profiles. Foams formulated with recycled polyols have a lower bulk density (88.3–96.9 kg m^−3^) and smaller cell sizes compared to a conventional reference RPUF. The addition of recycled polyols (up to 10 wt.%) into the formulation causes a slight decrease in compressive properties, whereas tensile strength and modulus values increase remarkably.

## 1. Introduction

Polyurethane (PU) is one of the most versatile polymers, offering a wide variety of commercial applications. PUs can be classified mainly into foams and CASEs (Coatings, Adhesives, Sealants, Elastomers) [1]. Foams are further subdivided into flexible foams, such as those used in mattresses, car seats or packaging; and rigid foams, which are generally used as insulation in buildings and in commercial and domestic refrigeration [2,3]. However, one of the main drawbacks to their great commercial success is the challenging management of the large amount of waste that is generated when the products including those foams reach their end of life [4]. Current environmental legislation and transition to a circular economy model point out an alternative to landfilling or disposal of polymer waste: recycling [5]. Polyurethane recycling processes can be divided into physical and chemical treatments. Physical recycling processes do not modify the internal structure of the polymer; instead, the polymer residues are mechanically processed into flakes, granules or powder to be used in the production of new materials [6,7]. These physical processes can be successfully applied to thermoplastic polymers but are ineffective for most types of polyurethane due to their thermosetting nature. Regarding chemical recycling processes, using thermochemical or solvolysis reactions, such as glycolysis [8,9,10], aminolysis [11,12,13], alcoholysis [14,15] or hydrolysis [16,17], polymers are broken down into basic hydrocarbon units or monomers that can be used in the chemical industry as raw materials. Glycolysis is undoubtedly the most important polyurethane material recovery process with remarkable advances in all different classes of polyurethane, including rigid foams [18,19,20,21,22,23,24]. The glycolysis of RPUFs (Equation (1)) consists of treating the residues with a low molecular mass glycol, thus obtaining a homogeneous, single-phase product with low viscosity and high hydroxyl value that can be used as a partial substitute for commercial polyether polyols in the synthesis of new rigid foams [25,26].
(1)$${R\text{-}\mathrm{NH}\text{-}\mathrm{CO}\text{-}O\text{-}R}^{\prime }+\mathrm{OH}\text{-}R^{\prime \prime }\text{-}\mathrm{OH}\;\to \;R\text{-}\mathrm{NH}\text{-}\mathrm{CO}\text{-}O\text{-}R^{\prime \prime }\text{-}\mathrm{OH}+\mathrm{OH}\text{-}R^{\prime }$$

Regarding prior research carried out on glycolysis of post-industrial waste RPUFs, Morooka et al. [18] chemically recycled RPUFs from refrigerators using diethylene glycol (DEG) as solvent and BaO or diethanolamine (DEA) as catalysts. They observed that the obtained glycolysate could be added to commercial polyol (up to 10 wt.%) to produce new RPUFs with thermal conductivities and compressive strengths similar to those of conventional foams. Zhu et al. [20] obtained higher yields by using ethylene glycol (EG) instead of DEG in the glycolysis of rigid foams from refrigerators. They observed a higher catalytic efficiency of NaOH and defined the optimal reaction conditions to be an EG:PU ratio of 1:1 by mass, a catalyst concentration of 1 wt.%, a temperature of 198 °C and a reaction time of 2 h. Recovered polyols were incorporated up to 10 wt.% with respect to the total amount of polyol in the new foam formulations.

Polyurethane chemistry is fundamentally based on the condensation reaction between polyols and diisocyanates. These isocyanate groups are very reactive to species with active hydrogens, such as hydroxyl groups, urethane groups and water. For this reason, during the foaming process of RPUFs, many exothermic reactions of isocyanate groups occur consecutively [27]. The main reaction that takes place is shown in Equation (2), known as a gelling reaction, in which the isocyanate groups and the hydroxyl groups of the polyol produce a cross-linked internal structure due to the urethane group that is generated.
(2)$${R\text{-}N=C=O+R}^{\prime }\text{-}\mathrm{OH}\;\to \;R\text{-}\mathrm{NH}\text{-}\mathrm{CO}\text{-}O\text{-}R^{\prime }$$

Equation (3) shows the reaction of isocyanate groups with water, known as blowing reaction. The isocyanate group reacts with water to form an amine and CO_2_. The gas expands and cells are formed in which the carbon dioxide is encapsulated. This reaction is highly exothermic and provides the main source of heat value required in the expansion and drying of the foam.
(3)$$R\text{-}N=C=O+H_{2}O\;\to \;R\text{-}{\mathrm{NH}}_{2}+{\mathrm{CO}}_{2}$$

Furthermore, isocyanate reacts additionally due to its high reactivity in the presence of protons. These reactions are known as crosslinking reactions. For instance, amines formed during the blowing reaction react with the free isocyanate, yielding substituted urea (Equation (4)).
(4)$$R\text{-}N=C=O+R^{\prime }\text{-}{\mathrm{NH}}_{2}\;\to \;R\text{-}\mathrm{NH}\text{-}\mathrm{CO}\text{-}\mathrm{NH}\text{-}R^{\prime }$$

Although previous research has identified glycolysis as one of the most suitable chemical recycling processes for PU recycling, applications developed from those findings mainly focus on clean flexible PU foams with known composition. Recovery of polyols is easier and higher purity can be achieved due to a low intensive post-treatment required by glycolysis products of flexible PU. In the case of rigid PU, the reaction product is a single phase, and for that reason, the recovered polyols are mixed with the solvent and with other chemical by-products of the reaction and compounds derived from the composition of the PU waste. Hence, it is necessary to concentrate the recovered polyols and validate their use in the synthesis of new PUs. Furthermore, the present work deals with real PU waste that is currently being generated and landfilled in large quantities, so this paper aims to present a technically feasible solution based on the principles of the circular economy, thus allowing the manufacture of new value-added polyurethanes and closing the cycle of PU material through the application of a chemical recycling process.

The purpose of the research is to understand the steps involved in the chemical recycling of post-industrial complex waste RPUFs for housing and bracket applications, and the synthesis of new recycled foams based on sustainable polyols. After purifying the reaction products by vacuum distillation, the recycled polyols have been incorporated into new formulations of RPUFs. The recovered polyols have been analyzed through various characterization techniques to examine their composition. Certain compounds, such as amines, that significantly influence the foaming process have also been identified and quantified. Foam synthesis reactions have been monitored, performing an analysis of the reactivity and exothermicity of the process. Temperature profiles and characteristic times of the reactions have been observed as a function of the amount of recycled polyol incorporated. Microscopic structures of foams obtained with recycled components have been compared to conventional foams and physical, chemical and mechanical properties of interest have been verified.

## 2. Materials and Methods

### 2.1. Materials

RPUFs that have been chemically recycled by glycolysis in this work are pieces for housing and bracket applications (70–80 kg m^−3^). They are industrially produced pieces, manufactured and provided by the company Arcesso Dynamics S.L. (Barcelona, Spain). Ethylene glycol (EG) and NaOH were used as the solvent and catalyst for glycolysis, respectively. For the synthesis of foams, Arcesso Dynamics S.L. provided a mixture of Lupranol^®^ 3300, Lupranol^®^ 3422 and Lupranol^®^ 3402, used as conventional commercial polyol (CP) with a hydroxyl value of 395 mg KOH g^−1^. A polymeric 4,4’-diphenylmethane diisocyanate (pMDI) with an isocyanate (NCO) content of 31% was also provided by Arcesso Dynamics S.L. Dimethylaminoethanol (DMAE) and distilled water were used as foaming catalyst and blowing agent, respectively. All chemicals were used as received.

### 2.2. Glycolysis of Waste RPUFs and Purification

Waste polyurethane foams were milled by means of a blade mill to particle sizes smaller than 2 mm in diameter prior to the glycolysis reaction. The selected glycol was fed into a 500 mL three necked glass reactor, equipped with a stirrer that was set at 100 rpm, a thermometer, and reflux condenser. Ethylene glycol as the glycolysis reagent and NaOH as the catalyst were added at different mass ratios given in Table 1 and preheated to the boiling temperature of the glycolysis reagent used. Finally, waste PU pieces were fed. Glycolysis conditions given in Table 1 are optimum values determined in previous works carried out by Gaiker to reach full depolymerization of PU waste. The final reaction product was filtered under pressure and distilled by means of vacuum distillation in a rotary evaporator, under a vacuum close to 50 hPa and an oil bath at 140 °C, for approximately 3 h and 30 min. Finally, two products were obtained: the distilled glycolysis agent and the recycled polyol, which were collected separately to analyze them and, in the case of the polyol, to use it in the synthesis of new RPUFs.

### 2.3. Synthesis of New RPUFs

New RPUFs were synthesized in 200 mL polypropylene cups by mixing different amounts of commercial polyol (CP), recycled polyol (RP), water (foaming agent), catalyst and isocyanate by a two-step method. First, the mixture of polyols, water and catalyst (component A) was stirred at 2000 rpm for a given time. Once a homogeneous mixture was obtained, the appropriate amount of pMDI (component B) was added, stirred for only 10 s and then free foaming was observed. Finally, the obtained foams were left to fully cure for at least 24 h at room temperature before performing the measurements. The NCO index was fixed at 1.1. The contents of the RPs with respect to total polyol weight were set at 0%, 5%, 10% and 15% in order to validate the inclusion of recycled polyols into new PU formulations. The RPUF prepared only with commercial polyol was designated as COM. The formulations of the RPUFs with different contents of RPs are summarized in Table 2.

### 2.4. Characterization of Polyols

Hydroxyl and acid values of the polyols were determined by titration methods according to ASTM E1899-16 and ASTM D4662 standards, respectively. The viscosity of the polyols was determined by means of an Anton Paar rotational rheometer, model MCR 501. Gel Permeation Chromatography (GPC) was used to determine the average molecular weight (M_n_) of the polyols. Measurements were performed with a Viscotek GPCmax VE-2001 TDA 302 Detector. The chemical structure of the products obtained was measured by means of a Fourier transform infrared (FTIR) spectrometer (Shimadzu IRAffinity-1S) in transmittance mode in the wavenumber range from 600 to 4000 cm^−1^ at a resolution of 4 cm^−1^. Thermogravimetric analysis (TGA) was carried out to determine the thermal stability of the materials by means of a Mettler-Toledo thermobalance, model TGA/DSC 1 Stare System. The analysis was conducted by heating the sample from room temperature to 600 °C, with a ramp of 20 °C min^−1^ under a constant N_2_ flow (50 mL min^−1^).

### 2.5. Characterization of RPUFs

To characterize the reactivity of the RPUFs on foaming and polymerization, characteristic times (cream time, gel time, rise time, tack-free time) of PU foam synthesis were recorded according to ASTM D7487-18, and temperature profiles were evaluated with the aid of a thermocouple. The cell structures of the RPUFs were studied using a scanning electron microscope (SEM, EVO50, ZEISS) at the accelerating voltage of 20 kV. The samples were coated with gold/palladium 80/20 wt.% before observation. Bulk densities of RPUFs were measured according to the ASTM D1895-17 standard. Tensile strength and compressive strength of the foams were determined according to the ASTM D790-10 and ASTM D1621-00 standards, respectively.

## 3. Results and Discussion

### 3.1. Analysis and Characteristics of the Polyols

Table 3 shows the results for hydroxyl, acid and average molecular weight values, along with the viscosity at 25 °C of the commercial and recycled polyols. RPs present significantly higher hydroxyl values than the commercial polyol because of the presence of glycolysis by-products and ethylene glycol (hydroxyl value of the EG: 1807.6 mg KOH g^−1^) remaining in the polyol after purification of the glycolysate. The presence of by-products can be explained by considering side reactions in the glycolysis process besides the transesterification reaction between PU and glycol, such as the glycolysis of urea groups producing a carbamate and an aromatic amine [3]. These urea groups are present in the PU structure due to the amines formed in the blowing reaction during the foaming process, which in turn react with the free isocyanate to produce substituted urea [27]. Although the hydroxyl values of the recycled polyols are high, they are within the range for polyols used in the synthesis of RPUFs [28].

The acid value is higher in RPs compared to the commercial polyol, although it does not exceed the maximum acidity (10 mg KOH g^−1^) that polyols should possess [28]. The acid value mainly indicates the concentration of carboxyl groups present in the polyols, therefore, a limitation of the glycolysis of waste RPUFs is observed [29]. RPs are much more viscous than commercial polyol, possibly due to the higher number of hydrogen bonds present and directly related to a higher hydroxyl and average molecular weight values of the RPs [30]. Although the viscosity of RPs is high, its value is similar to the ones of commercial polyols used in the synthesis of RPUFs [31]. It must be noted that average molecular weight (M_n_) values of the RPs are higher compared to the commercial polyol, directly related to their higher viscosity and hydroxyl functionality.

Figure 1 shows the FTIR spectra of the glycolysis products compared to the corresponding RPs after distillation. A reduction in the intensity of the peak at 3345 cm^−1^ (corresponding to O-H bond) was observed in the polyols after purification by vacuum distillation. The peak at 1614 cm^−1^ indicates the presence of amines in the RPs, a by-product of PU glycolysis due to the urea groups present in the polyurethane structure. These amines could accelerate the synthesis reaction of new rigid foams, leading to undesired reactions during foaming.

All polyols show absorption peaks corresponding to the stretching vibrations of the hydroxyl groups (O-H) around 3345–3420 cm^−1^, and sp^3^ stretching vibration of the C-H bond (2870 and 2968 cm^−1^). Absorption peaks at 1373 and 1454 cm^−1^ correspond to bending vibrations of the methyl (-CH_3_) bond in the polyol chain, while the most prominent characteristic peak is the one at 1082 cm^−1^ corresponding to stretching vibrations of the C-O-C group. However, FTIR spectra of RPs present some differences compared to the CP as seen in Figure 2. Characteristic peaks at 1737 cm^−1^ corresponding to the C=O bond of the urethane and urea groups and bending vibrations of the N-H bond (1514 and 1614 cm^−1^) are observed due to the limitation of the PU glycolysis reaction. The high intensity of the characteristic peak of the latter bond could also indicate the presence of amines in the RPs.

Distilled products were also analyzed and compared with EG (Figure 3), the glycolysis agent of depolymerization. It is observed that FTIR spectra of the distillates are practically identical to the EG spectra, although the peak at 3345 cm^−1^ in RP2 distillate was slightly more intense due to possible impurities.

Thermogravimetric analysis was performed to study the thermal stability of polyols. Figure 4a shows the degradation range (200–400 °C) of the commercial polyol, which is smaller than the RPs range (130–500 °C). In other words, the commercial polyol seems to present higher thermal stability compared to the RPs recovered via glycolysis. The DTG curve indicates the temperature at which the rate of thermal degradation is the highest (Figure 4b). For the CP, this temperature is about 320 °C, while for RP1 is 280 °C and for RP2 is 245 °C. This phenomenon may be a consequence of the relatively low thermal stability of urethane groups, which are present in the RPs because of the limitation of glycolysis. After thermal degradation, a higher percentage of solid residue was remaining in RPs compared to the CP. Solid residue in RP1 and RP2 represented 13% and 11%, respectively, whereas the undegraded solid did not exceed 1% for the CP. These results could be due to the high concentration of hydroxyl groups in the RPs.

### 3.2. Synthesis of PU Foams Based on Recycled Polyols

The absence of catalyst in formulations of RPUFs is remarkable (Table 2) since the amines present in the RPs accelerate the reaction without the need to add the additive. The physical appearance and internal structure of the obtained foams are shown in Figure 5. A greater cross-linking is observed in the structure of the foams as the amount of RP incorporated increases.

Figure 6 shows the temperature profiles registered during the synthesis of RPUFs with both RPs. More exothermic profiles are observed as the amount of RP incorporated in the formulations increases, due to crosslinking reactions between the amines present in the recovered polyols and the isocyanate producing substituted urea [27]. This particular reaction at room temperature in the absence of catalyst is much faster than the reaction with alcohols, a conclusion that the reduced characteristic times of the reactions also confirm. However, it should be noted that reactions with 5 wt.% and 10 wt.% of RP2 are not as exothermic as reactions when the same amount of RP1 is added.

Characteristic times for each RPUF synthesis are compared in Figure 7. Remarkable differences are observed when RPs are added to the formulation. As previously discussed, foaming is accelerated when both RPs are incorporated due to the possible presence of amines. As the content of the RP increases the characteristic times become shorter, even the tack-free time is shorter than the rise time for 10 wt.% and 15 wt.% formulations of both RPs. This phenomenon can be attributed to the catalytic effect of the amines present in the RPs. Although these time values are really short, RPUFs with 5 wt.% RP in their formulation present characteristic times within the range of the foaming process.

Figure 8 shows different SEM images obtained from RPUFs synthesized with different types and contents of RPs (Table 2). All samples present a polyhedral and closed-cell structure. Cell size decreases and the heterogeneity of the sample structure increases while RP content is increased due to the fast reactivity and high viscosity of the RPs. Cell size range of the RPUF designated as COM is between 219–547 μm, significantly higher values than those of the samples containing recycled polyols, designated as RP1-5 (180–532 μm), RP1-10 (155–387 μm) and RP1-15 (132–349 μm) for samples synthesized with RP1, and RP2-5 (162–393 μm), RP2-10 (135–376 μm) and RP2-15 (118–362 μm) for samples synthesized with RP2. Thus, higher reactivity and viscosity of the RPs caused an acceleration of foaming and influenced the smaller cell size of the samples while increasing the RP content [32]. As previously discussed, recovered polyols are much more viscous than the commercial polyol, so the CO_2_ generated during the blowing reaction (Equation (3)) gets trapped in the foam structure. This causes a greater expansion of the PU sample, reaching lower bulk density values as the amount of recycled polyol is increased.

The mechanical properties of the synthesized foams are evaluated just for RP1 since both recycled polyols present similar characteristics in terms of reactivity, viscosity, foaming parameters and chemical structure. RPUFs synthetized with RP1 present lower bulk density values (88.3 ± 0.75 kg m^−3^ for RP1-10 and 96.9 ± 0.92 kg m^−3^ for RP1-5) than those of the conventional foam (102.5 ± 0.84 kg m^−3^). The introduction of RP1 in the formulation of RPUFs involves increasing the molecular mass of the polyol mixture, resulting in lower bulk density foams [32,33]. Tensile properties improved notably when RP1 was added to the formulation of new RPUFs. The tensile strength value of the commercial foam was 0.91 ± 0.05 MPa, whereas for foams with 5 wt.% and 10 wt.% of RP1 were 1.43 ± 0.04 MPa and 1.04 ± 0.04 MPa, respectively (Figure 9). Equally, tensile modulus values were higher when RP1 was added to the formulation. An increase in the RP content leads to an improvement of tensile properties of RPUFs, probably due to a higher cross-linking of the foams with recycled components [34]. Partial substitution of commercial polyol by RP1 up to 10 wt.% caused a slight decrease in the compressive strength values.

## 4. Conclusions

New RPUFs were prepared using two types of RPs with different properties. The RPs were obtained via glycolysis of post-industrial waste polyurethane foams and subsequent purification by vacuum distillation. The RPs were mixed with conventional polyether polyol to prepare RPUFs.

Recycled polyols presented significantly higher hydroxyl, acid, average molecular weight and viscosity values than the commercial polyol because of the presence of glycolysis by-products (carbamates, amines) and ethylene glycol remaining in the polyol after purification of the glycolysate. FTIR analysis concluded that RPs present C=O (1737 cm^−1^) and N-H (1514 and 1614 cm^−1^) bonds of urethane and urea groups due to the limitation of the PU glycolysis reaction. It is possible that the N-H bond indicates the presence of amines, a by-product of PU glycolysis due to the urea groups present in the polyurethane structure.

TGA determined that RPs exhibit lower thermal stability since their degradation range is much wider (130–500 °C) than the commercial polyol range (200–400 °C) due to the low thermal stability of urethane groups, which are present in the RPs because of the limitation of the glycolysis reaction.

The amines that remained in the RPs promoted the reactivity, judging by the values obtained for characteristic times and more exothermic temperature profiles during the foaming. Higher reactivity and viscosity of the RPs caused an acceleration of the foaming reaction and influenced smaller cell size according to SEM images.

Bulk density and compressive strength values of the RPUFs decreased slightly with the incorporation of 5 wt.% and 10 wt.% of RP1. However, the tensile properties of RPUFs were remarkably increased with the addition of recycled polyols.

All in all, prepared RPUFs show an improvement of tensile properties when recycled polyols are introduced, probably as a consequence of a higher cross-linking of the samples with recycled components. This is relevant due to the housing and bracket applications that the RPUFs are used for, being an improvement over conventional foams. Nevertheless, the inclusion of recycled polyols into prepared RPUFs is still a small percentage, since the reactivity of the RPs is too high to introduce more than 15 wt.% of recovered polyol into the formulations. Incorporation percentages are usually lower in consulted papers regarding recycling of RPUFs because the polyols are mixed with other chemical species and their intense purification is technically difficult. Therefore, the inclusion of more than 10 wt.% of recovered polyol in the final PU formulation with adequate properties can be considered an interesting and significant advance in the field of chemical recycling of rigid PU waste. In addition, future research to be carried out by our research group will modify and optimize the formulation of the foams synthetized with RPs, introducing a less reactive isocyanate to the formulation of the foams, thus enabling the incorporation of higher concentrations of polyols coming from the polyurethane wastes.

## Figures and Tables

**Figure 1 polymers-14-01157-f001:**
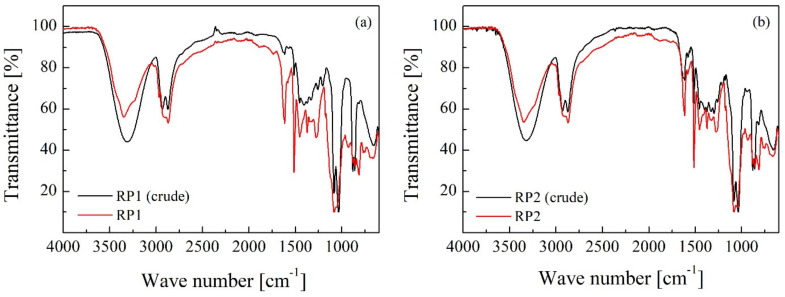
FTIR spectra of (**a**) crude and distilled RP1 and (**b**) crude and distilled RP2.

**Figure 2 polymers-14-01157-f002:**
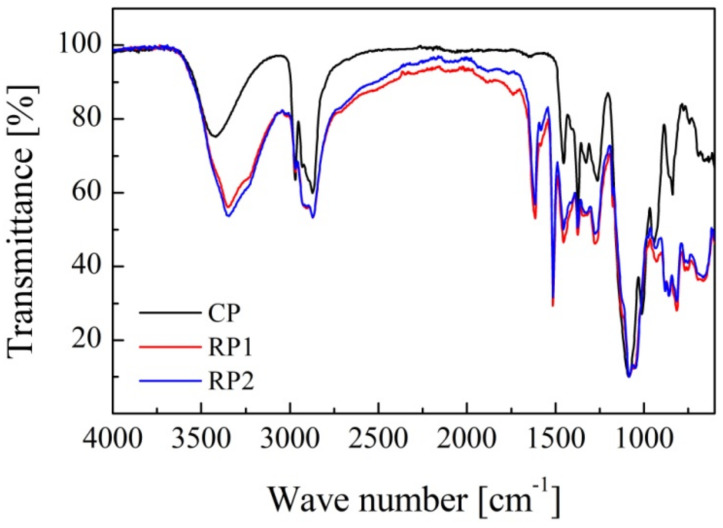
FTIR spectra of RPs and commercial polyol.

**Figure 3 polymers-14-01157-f003:**
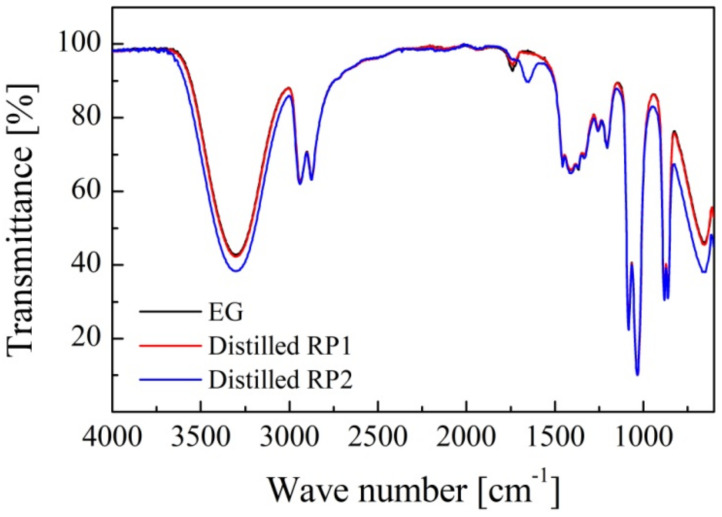
FTIR spectra of EG and distilled solvents.

**Figure 4 polymers-14-01157-f004:**
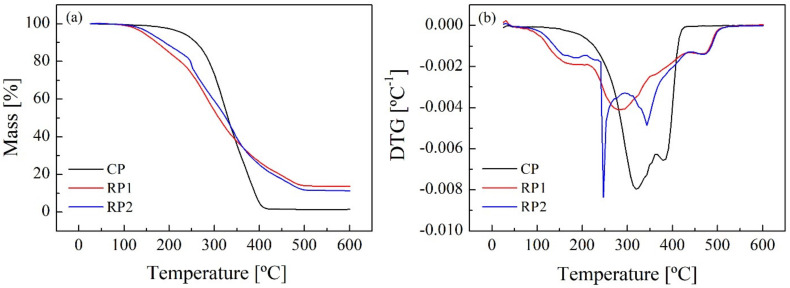
Thermogravimetric TGA (**a**) and DTG (**b**) curves of the polyols.

**Figure 5 polymers-14-01157-f005:**
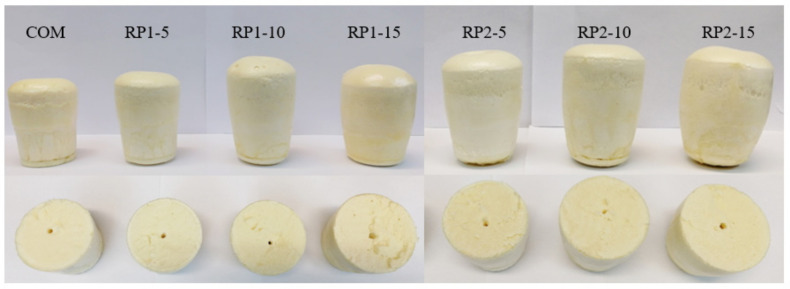
Physical appearance and internal structure of RPUFs synthesized with RPs.

**Figure 6 polymers-14-01157-f006:**
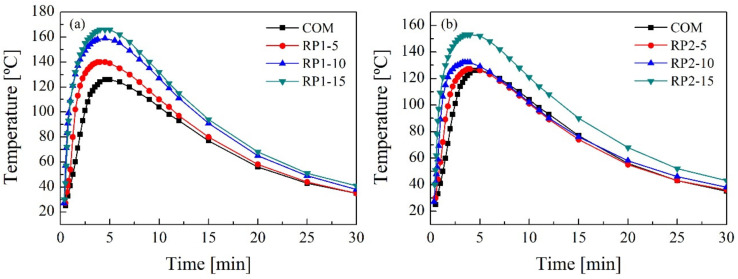
Temperature profiles in reactions with (**a**) RP1 and (**b**) RP2.

**Figure 7 polymers-14-01157-f007:**
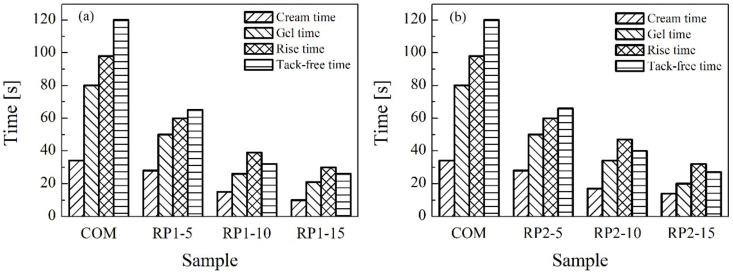
Characteristic times in reactions with (**a**) RP1 and (**b**) RP2.

**Figure 8 polymers-14-01157-f008:**
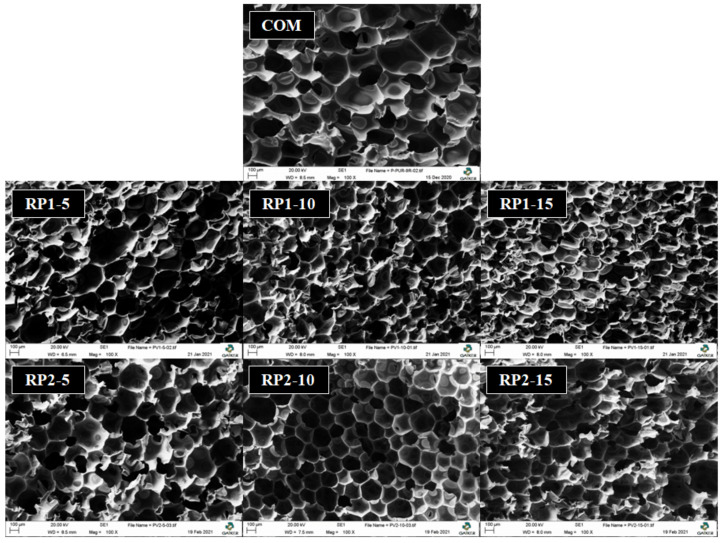
SEM images of cross sections of RPUFs with different RP contents.

**Figure 9 polymers-14-01157-f009:**
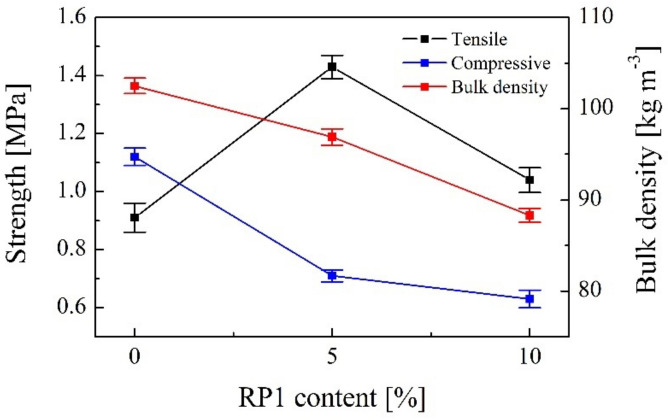
Tensile strength, compressive strength and bulk density according to RP1 content.

**Table 1 polymers-14-01157-t001:** Conditions of the glycolysis reactions.

Recycled Polyol Code	Solvent	Catalyst	Solvent:PU [g:g]	Catalyst:PU [mol:g]	Temperature [°C]	Time [h]
RP1	EG	NaOH	4:1	0.002:1	198	2
RP2	EG	NaOH	1.5:1	0.002:1	180	2

**Table 2 polymers-14-01157-t002:** Sample codes and formulations for RPUFs.

Sample	COM	RP1-5	RP1-10	RP1-15	RP2-5	RP2-10	RP2-15
Component A [php] ^1^							
CP	100	95	90	85	95	90	85
RP1	-	5	10	15	-	-	-
RP2	-	-	-	-	5	10	15
Water	1	1	1	1	1	1	1
DMAE	1	-	-	-	-	-	-
Component B							
NCO index	1.1	1.1	1.1	1.1	1.1	1.1	1.1

^1^ per hundred polyol.

**Table 3 polymers-14-01157-t003:** Characteristics of the commercial and the recycled polyols.

Polyol Code	Hydroxyl Value [mg KOH g^−1^]	Acid Value [mg KOH g^−1^]	M_n_ (g/mol)	M_w_/M_n_	Viscosity @25 °C [Pa·s]
RP1	940 ± 85	8.7 ± 0.09	873.8 ± 32	1.031 ± 0.04	17,820 ± 970
RP2	1130 ± 95	4.48 ± 0.42	885.4 ± 35	1.037 ± 0.04	10,720 ± 610
CP	395 ± 65	0.09 ± 0.02	535.5 ± 21	1.014 ± 0.04	1690 ± 35

## Data Availability

The data presented in this study are available on request from the corresponding author.

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
