# Peer review of "Synthesis of Rigid Polyurethane Foams Incorporating Polyols from Chemical Recycling of Post-Industrial Waste Polyurethane Foams"

_polymers, 2022, doi:10.3390/polym14061157_

Round 1

Reviewer 1 Report

Thank you for the improvement

Author Response

"Thank you for the improvement"

The authors would like to thank the reviewer for his valuable contributions that have significantly improved the manuscript

Reviewer 2 Report

My previous comment was:

The manuscript lacks novelty. Glycolysis of PU is widely known and manuscript is not presenting any new information about this process. Therefore, either manuscript should be significantly enhanced and reviewed keeping in mind current state of the art or should be rejected. 

Authors provided some explanation in non-published material, but the paper remained unchanged considering this issue. Introduction does not indicate that presented paper is novel in any way. It has to be corrected, because right now seems that Authors do not care about the comment. 

Authors also claimed that novelty is that "Scientific papers about complex post-consumer wastes are less abundant and they usually only cover up to the recovery of polyols without further research on the options for inclusion back into PU formulation. ". But there are multiple works on using polyols from glycolysis in manufacturing of PU. What is the difference if there is 1 paper covering glycolysis and manufacturing of PU or 2 separate papers describing both steps more comprehensively? I see totally no novelty here.

I did not go to the results, but some suggestionns about the Experimental part:

Authors are writing that the novelty is they used actual waste. But it was not characterized in any manner except for the density, which is not enough. 

Table 1 presents some conditions of glycolysis, but why only two samples were analysed? Why for example no solvent:PU ratio of 1.5:1 and temperature of 198 degrees? No explanation for that, values out of nowhere.

Why only 15% of polyol from recycling was used? It makes hardly any sense, the proportions should be reversed.

And finally:

Finally, authors would like to point out that results presented in this manuscript represent a real application that will be implemented in a pilot plant scale in the framework of the FOAM2FOAM Project, funded by the Spanish Ministry of Science and Innovation.

The implementation in the industry is not equal to scientific novelty.

Reviewer 3 Report

Preparation and characteristics of rigid polyurethane foams (RPUFs) synthesized from polyols obtained by glycolysis of post-industrial waste RPUFs are described in the manuscript.  The paper is interesting and could be published after revision.

-Introduction of the paper is rather long and also with well known information. It seem that only research in field of glycolysis of post-industrial waste of RPUFs should be presented in the introduction ?

-What is molecular weight of the obtained polyols. GPC analyses would demonstrate mass distribution in the polyols ?

-Why the RPs were mixed with conventional polyether polyol to prepare RPUFs ?

- What would be an optimal wt.% of RP for preparation of rigid polyurethane foams (RPUFs) ?

-The authors should describe clearly advantages and disadvantages of the prepared RPUFs as compared with those of other polyuretanes used in this field of applications. The advantages and disadvantages should be exactly presented in conclusions.

Round 2

Reviewer 2 Report

"“Authors are writing that the novelty is they used actual waste. But it was not characterized in any manner except for the density, which is not enough.”

In agreement with the reviewer’s remark regarding the characterization of the PU waste, authors would like to note that our research group is preparing another manuscript dealing with the complete characterization and catalytic glycolysis process of waste PU in more detail."

Great, but currently this paper is reviewed. Please see the instructions for Authors of Polymers (or any other) journal:

  • Materials and Methods: They should be described with sufficient detail to allow others to replicate and build on published results. New methods and protocols should be described in detail while well-established methods can be briefly described and appropriately cited. Give the name and version of any software used and make clear whether computer code used is available. Include any pre-registration codes.

Therefore, at least some characterization of applied PU foams have to be presented. Without it the paper cannot be accepted. 

"

“Table 1 presents some conditions of glycolysis, but why only two samples were analysed? Why for example no solvent:PU ratio of 1.5:1 and temperature of 198 degrees? No explanation for that, values out of nowhere.”

Glycolysis conditions given in Table 1 are optimum values determined in previous works carried out by Gaiker to reach full depolymerization of PU waste. Following the recommendation of the reviewer, this explanation was introduced in the re-submitted manuscript."

So please present these results or at least provide the reference.

"

“Why only 15% of polyol from recycling was used? It makes hardly any sense, the proportions should be reversed.”

We started blending small amounts of recycled polyols with the commercial one, until reactivity of the synthesis was too fast and characteristic times were too short to increase the amount of recycled polyol into the formulation. Prior research includes higher percentages of recovered polyols, but in most cases clean and pre-consumer flexible PU wastes are used. Consequently, recovery of polyols is easier and higher purity can be achieved. Regarding research with rigid PU waste, incorporation percentages are usually lower because the polyols are mixed with other chemical species and their intense purification is technically difficult when process scale-up factors are considered. Therefore, the inclusion of more than 10% of recovered polyols in the final PU formulation with adequate properties can be considered an interesting and significant advance in the field of chemical recycling of rigid PU waste."

Then please provide explanation in paper and cite proper articles, which use such a low contents.

Reviewer 3 Report

If editor and other reviewers agree that the paper is suitable for “Polymers” I recommend the paper for publication after the revision.

This manuscript is a resubmission of an earlier submission. The following is a list of the peer review reports and author responses from that submission.

Round 1

Reviewer 1 Report

The article does not match the content of the Special Issue "Development in Fiber-Reinforced Polymer Composites." 

QUESTIONS AND REMARKS:

What do you estimate as the scientific or technical novelty of this work?

Line 2, the title: "recycled polyols" REMARK: The polyols were not recycled. Waste polyurethane foams were recycled. CORRECT: 

The obtained foams were not based on recycled polyols from chemical recycling of post-industrial waste polyurethane foams but were based on polyol mixtures containing these polyols with their highest content of 15 wt.%.

BETTER Synthesis of rigid polyurethane foams incorporating polyols from chemical recycling of post-industrial waste polyurethane foams.

LIne 98: They are industrially produced pieces, manufactured and provided by the company Arcesso Dynamics S.L REMARK What application were these elements for? Their density shows they were not for thermal insulation.

Line 99  A conventional commercial polyol (CP) REMARK: What is the commercial name of this polyol?

Line 93 REMARK: In the materials section information about the glycol used for glycolysis is missing. (Table 1: EG - ethylene glycol) 

Line 109: Glycolysis reagent and catalyst were added REMARK: chemical names missing

Line 118: 2.3. Synthesis of new RPUFs. REMARK: the amine reactivity was not estimated and therefore not considered in the formulation shown in Table 2. The amine functionality could be transformed into hydroxyl functionality, as described in many publications.

Line 124 foams were left to dry REMARK left to fully cure (there is no drying!)

LIne 268: Figure 11. Tensile strength, compressive strength, and bulk density according to RP content - REMARK: Error bars missing

LIne 292 Consequently, the results confirm that RPs obtained via glycolysis of waste polyurEthane foams represent favourable alternatives for the preparation of RPUFs with quality performance. REMARK: How could it be a favorable alternative if only a small amount of virgin polyol was replaced?

Reviewer 2 Report

The manuscript lacks novelty. Glycolysis of PU is widely known and manuscript is not presenting any new information about this process. Therefore, either manuscript should be significantly enhanced and reviewed keeping in mind current state of the art or should be rejected. 

Reviewer 3 Report

This paper presents an exciting study on developing PU foams using recycled PU. The authors have investigated the various parameters for the glycolysis process to obtain polyol from recycled PU. I have the following comments that should be addressed to improve this paper:

  1. The authors have made samples by blending recycled polyol with commercial polyol. Can you make a sample with 100% recycled polyol for comparison purposes?
  2. How did you decide the ratios of blending of RP and CP?
  3. Please mention the ASTM standards for tensile and compressive testing
  4. Density is a good indicator of the strength properties of foam. Please do the density measurements. Please refer the following paper to know more details on the impact of density on various properties of foam: https://doi.org/10.1007/s10924-019-01477-0 
  5. Please combine figures 4, 5 and 6. It will be easy to compare the results. 
  6. Please mention the cell size of the various samples shown in SEM images. 
  7. Did you use any surfactant in the fabrication of samples? Surfactants can have a big impact on the cell size. Please refer the following paper showing the impact of cell size on the tensile and compressive properties: https://doi.org/10.1177/0021955X20912200
  8. Figure 11: Is this fig showing results for RP1 or RP2?
  9. Why is tensile strength for 5% RP increasing and then decreasing for 10%?